# Hypercontractile Cardiac Phenotype in Mice with Migraine-Associated Mutation in the Na^+^,K^+^-ATPase α_2_-Isoform

**DOI:** 10.3390/cells12081108

**Published:** 2023-04-07

**Authors:** Rajkumar Rajanathan, Clàudia Vilaseca i Riera, Tina Myhre Pedersen, Christian Staehr, Elena V. Bouzinova, Jens Randel Nyengaard, Morten B. Thomsen, Hans Erik Bøtker, Vladimir V. Matchkov

**Affiliations:** 1Department of Biomedicine, Aarhus University, 8000 Aarhus, Denmark; 2Department of Basic Science, School of Medicine and Health Sciences, International University of Catalonia, 08195 Barcelona, Spain; 3Department of Clinical Medicine, Core Center for Molecular Morphology, Section for Stereology and Microscopy, Aarhus University, 8000 Aarhus, Denmark; 4Department of Pathology, Aarhus University Hospital, 8200 Aarhus, Denmark; 5Biomedical Sciences, University of Copenhagen, 1168 Copenhagen, Denmark; 6Department of Cardiology, Aarhus University Hospital, 8200 Aarhus, Denmark

**Keywords:** Na^+^,K^+^-ATPase, α_2_-isoform, contractility, relaxation, ouabain, Langendorff, perfusion, cardiac function, familial hemiplegic migraine, cardiovascular

## Abstract

Two α-isoforms of the Na^+^,K^+^-ATPase (α_1_ and α_2_) are expressed in the cardiovascular system, and it is unclear which isoform is the preferential regulator of contractility. Mice heterozygous for the familial hemiplegic migraine type 2 (FHM2) associated mutation in the α_2_-isoform (G301R; α_2_^+/G301R^ mice) have decreased expression of cardiac α_2_-isoform but elevated expression of the α_1_-isoform. We aimed to investigate the contribution of the α_2_-isoform function to the cardiac phenotype of α_2_^+/G301R^ hearts. We hypothesized that α_2_^+/G301R^ hearts exhibit greater contractility due to reduced expression of cardiac α_2_-isoform. Variables for contractility and relaxation of isolated hearts were assessed in the Langendorff system without and in the presence of ouabain (1 µM). Atrial pacing was performed to investigate rate-dependent changes. The α_2_^+/G301R^ hearts displayed greater contractility than WT hearts during sinus rhythm, which was rate-dependent. The inotropic effect of ouabain was more augmented in α_2_^+/G301R^ hearts than in WT hearts during sinus rhythm and atrial pacing. In conclusion, cardiac contractility was greater in α_2_^+/G301R^ hearts than in WT hearts under resting conditions. The inotropic effect of ouabain was rate-independent and enhanced in α_2_^+/G301R^ hearts, which was associated with increased systolic work.

## 1. Introduction

Cardiovascular diseases are the leading cause of global mortality, often as a result of cardiac dysfunction and failure [1]. In 1628, William Harvey described the function of the heart as a pump providing the kinetic energy necessary for blood flow through the circulation [2]. The amount of blood pumped by the heart is determined by heart rate, preload, afterload, and the force with which the heart contracts, i.e., cardiac contractility [3]. One of the crucial mediators of cardiac contraction is intracellular Ca^2+^ ions [4]. During the action potential of a cardiomyocyte, Ca^2+^ influx through the voltage-gated Ca^2+^ channels increases [4]. These Ca^2+^ ions interact with and activate the ryanodine receptors on the sarcoplasmic reticulum leading to the release of Ca^2+^ ions, i.e., Ca^2+^-induced Ca^2+^ release [4]. The increase in intracellular Ca^2+^ concentration ([Ca^2+^]_in_) facilitates the binding of Ca^2+^ ions to troponin C, impeding its function, which allows the interaction of myosin and actin filaments with consequent contraction of the contractile machinery [4]. Hence, to a large extent, the regulation of [Ca^2+^]_in_ or the sensitivity of the contractile machinery to [Ca^2+^]_in_ defines cardiac contractility [4]. Understanding the underlying mechanisms regulating cardiomyocyte Ca^2+^ handling and, thus, cardiac contractility is important for the development of therapeutic strategies and clinical alleviation of cardiac diseases [5].

The Na^+^,K^+^-ATPase extrudes three Na^+^ ions across the cell membrane, and in return, it pumps two K^+^ ions into the intracellular space [6]. This electrogenic ion transport is important for many cellular functions, including the secondary transport of ions and substrates across the cell membrane, e.g., driving the function of the Na^+^/Ca^2+^ exchanger (NCX) [7,8]. The NCX is a bidirectional transporter for Ca^2+^ ions [9]. Under physiological conditions during diastole, the NCX uses the electrochemical gradient of the Na^+^ ions to extrude Ca^2+^ ions out of the cell with a stoichiometry of 3 Na^+^:1 Ca^2+^ [9]. However, the reversal potential for NCX transport in its conventional configuration is close to the resting membrane potential [10,11]. Therefore, Ca^+^ entry through the NCX is transiently favored during the cardiomyocyte action potential [10,11]. In addition to membrane depolarization, minor changes in the Na^+^ and Ca^2+^ concentrations may also change the direction of the Na^+^/Ca^2+^ transport [9].

Of the four known enzymatic and ion-transporting α-isoforms of the Na^+^,K^+^-ATPase, the α_1_- and α_2_-isoforms are expressed in muscle tissues, including cardiac muscle [12]. Whether the NCX-associated modulation of cardiac contractility is a preferential function of the α_1_- or α_2_-isoform is not fully elucidated and remains a controversial topic [8,13]. The α_2_-isoform has been described to mainly be localized in the transverse tubule (t-tubule) system [14,15]. Accordingly, a functional microdomain for [Ca^2+^]_in_ regulation comprised of the α_2_-isoform and the NCX localized in the close proximity of the t-tubule system to the sarcoplasmic reticulum has been proposed [8]. Based on this, it has been hypothesized that reduced abundance or activity of the α_2_-isoform, which is colocalized with the NCX, may modify the intracellular Na^+^ concentration and, thus, the NCX function resulting in increased [Ca^2+^]_in_ and augmented contraction [9]. This view suggests that the α_2_-isoform is a regulator of cardiac contractility, while the ubiquitously expressed α_1_-isoform has been suggested to perform a “housekeeping role”, regulating the bulk cytosolic Na^+^ concentration [16]. However, in rodents, the α_1_-isoform has been shown to be similarly or more abundantly expressed in the t-tubules than the α_2_-isoform [14,15]. Additionally, other studies have demonstrated the colocalization of the α_1_-isoform with the NCX in the t-tubules and its importance for cardiac contractility, questioning the specialized role of the α_2_-isoform [8,13,17]. Thus, more research is required to clarify the isoform-specific functions for cardiac contractility [13].

Ouabain is a cardiac glycoside with positive inotropic effects via the direct inhibition of the Na^+^,K^+^-ATPase function [18,19]. In rodents, ouabain inhibits the different isoforms of the Na^+^,K^+^-ATPase in a concentration-dependent manner [20]. At low concentrations (0.1–10 µM), ouabain inhibits the function of the α_2_-isoform with no effect on the α_1_-isoform [21]. Low concentrations of ouabain may, therefore, allow for the distinction between primary α_2_-isoform functions and α_1_-isoform functions [22].

Familial hemiplegic migraine type 2 (FHM2) is a condition associated with mutations in the *ATP1A2* gene encoding the α_2_-isoform of the Na^+^,K^+^-ATPase [23]. It has been shown that the FHM2-associated mutation, G301R, leads to both impediment of protein translocation to the cell membrane and functional disarray of expressed mutated protein [24,25]. We have demonstrated that mice heterozygous for the FHM2-associated mutation (G301R; α_2_^+/G301R^) are characterized by decreased expression of the α_2_-isoform but increased expression of the α_1_-isoform in their hearts [26]. This inverse expression of the α_1_- and the α_2_-isoforms in the α_2_^+/G301R^ mouse model offers a unique opportunity for studying the isoform-specific Na^+^,K^+^-ATPase contribution to cardiac contractility in a disease model relevant to human health. We have previously demonstrated that aged α_2_^+/G301R^ mice (>8 months old) have an augmented pro-contractile vascular response in vivo to ouabain at concentrations that inhibit the α_2_-isoform without any significant effect on the α_1_-isoform [27]. We proposed that this enhanced vascular response is due to fewer α_2_-isoform proteins expressed in the cell membrane [27]. This will yield a higher fractional occupancy of the α_2_-isoform in relation to a given ouabain concentration and potentiate the effect of ouabain [28,29]. Whether cardiac tissue from aged α_2_^+/G301R^ mice exhibits a similar augmented response to ouabain has not yet been addressed.

In the current study, we aimed to discern the isolated contribution of the α_2_-isoform to cardiac function in α_2_^+/G301R^ mice using low concentrations of ouabain (1 µM) that inhibit the α_2_-isoform and not the α_1_-isoform. The primary outcome of this study was cardiac contractility; however, we also investigated variables for relaxation, systolic work, and systolic duration. We hypothesized that isolated hearts from α_2_^+/G301R^ mice are more contractile and display a more augmented pro-contractile effect of ouabain than the hearts from wild-type (WT) mice, due to the reduced abundance of cardiac α_2_-isoform. Accordingly, we found that the reduced α_2_-isoform expression in isolated α_2_^+/G301R^ hearts was associated with rate-dependent enhancement of contractility when compared with WT hearts at baseline. In the presence of ouabain, systolic work increased in α_2_^+/G301R^ hearts, whereas systolic work remained unchanged for WT hearts. Moreover, cardiac contractility was greater in α_2_^+/G301R^ hearts than in WT hearts during atrial pacing in the presence of ouabain, which may suggest an augmented effect of ouabain in α_2_^+/G301R^ hearts.

This study demonstrates the significant role of the α_2_-isoform in the regulation of cardiac contractility in a murine model relevant to human disease, i.e., FHM2. We suggest that the chronic augmented cardiac contractility of α_2_^+/G301R^ hearts may be implicated in the development of cardiometabolic abnormalities described in aged α_2_^+/G301R^ mice [26].

## 2. Materials and Methods

### 2.1. Experimental Animals

A previous investigation suggested a changed cardiac phenotype in aged mice but not in young α_2_^+/G301R^ mice, and no difference between female and male mice of the same genotype was reported [26]. Due to their bigger body sizes, aged α_2_^+/G301R^ male mice were previously studied in vivo [27]. Therefore, to establish congruence with the previous study [27], the α_2_^+/G301R^ and WT male mice with a C57BL/6J background and above 8 months of age were used in the present study. Investigators were blinded regarding the genotype of animals throughout the experimentation and analysis. The α_2_^+/G301R^ mouse line was kept and bred at Aarhus University, Aarhus, Denmark, and it was generated as previously described [30]. Only heterozygous mice were available, since the homozygous genotype for the G301R mutation is lethal [31,32]. For the Langendorff experiments, seven α_2_^+/G301R^ and seven WT mice were euthanized, and their isolated hearts were exposed to ouabain (Sigma– Aldrich, Merck Life Science A/S, Søborg, Denmark). Four additional WT mice were used for time-control experiments, and their hearts were not exposed to ouabain. For stereological assessment, five α_2_^+/G301R^ mice and six WT mice were used. Mice were housed in rooms with controlled temperature (21.5 °C) and humidity (55%) with a 12:12 h light-dark cycle, and they had free access to food and water. All animal experiments were conducted according to the guidelines from Directive 2010/63/EU of the European Parliament on the protection of animals used for scientific purposes. The experimental protocol was approved by the Animal Experiments Inspectorate of the Danish Ministry of Environment and Food and reported in accordance with the ARRIVE (Animal Research: Reporting in vivo Experiments) guidelines [33].

### 2.2. Isolated Cardiac Perfusions

Isolated hearts were mounted in the Langendorff setup for functional assessment [34]. Briefly, mice were injected with heparin and pentobarbital i.p. (125 IU + 320 mg/kg, Leo Pharma, Denmark, and Virbac, Denmark, respectively) prior to the excision of their hearts and subsequent euthanasia by exsanguination. The aorta was cannulated (18 G) and the heart was perfused with Krebs–Henseleit buffer (Appendix A) bubbled with carbogen (95% O_2_ and 5% CO_2_). A custom-made fluid-filled balloon connected to a pressure transducer (ADInstruments Ltd., Oxford, UK) was then inserted into the left ventricle. The balloon was inflated to achieve a standard end-diastolic pressure of 5–10 mmHg, ensuring proper filling and the standardization of pressures within the ventricle. Coronary perfusion pressure was maintained at 60 mmHg in the aorta and 0 mmHg on the site of outflow. The temperature was maintained at approximately 37 °C throughout. Bipolar pacing electrodes connected to a stimulator unit (Danish Myo Technology A/S (DMT), Denmark) were placed on the right atrium. The electrical pulse width was set to 0.1 ms with a current amplitude of 10–30 mA. The excitation threshold was determined by gradually increasing current amplitude until cardiac capture.

Instrumentation of all 18 hearts was successful. One WT heart exposed to ouabain was excluded due to non-physiological pressure development (<25 mmHg). Three atrial pacing sessions were performed, consisting of a “warm-up” session and two subsequent sessions with measurements of cardiac variables (Figure 1). When pressure and intrinsic heart rate were stable and baseline recordings were made, atrial pacing was commenced at 10.5 Hz (630 BPM) and maintained for 30 s. After this initial “warm-up” pacing and a 5 min stabilization period, a second pacing was conducted in a similar manner. After the first two pacing sessions and 5 min stabilization, ouabain was added to the perfusate (final concentration of 1 µM). Ten minutes after ouabain was added to the perfusate, a final pacing session was performed in a fashion similar to that of the first two sessions. This time frame was chosen based on previous experimentation [27]. For time-control experiments, four WT hearts were exposed to the same regimen, but without ouabain.

### 2.3. Data Acquisition and Calculations

In the Langendorff experiments, a Power lab unit (PowerLab 8/35, ADInstruments, Dunedin, New Zealand) was used to collect cardiac variables with a sampling rate of 1 kHz, and the variables were recorded in LabChart Pro 8 (ADInstruments). Data were acquired from periods of 10 s during (i) baseline sinus rhythm, (ii) the first atrial pacing at 10.5 Hz (630 BPM) without ouabain, (iii) sinus rhythm after the first atrial pacing without ouabain, (iv) sinus rhythm in the presence of ouabain, and (v) the second atrial pacing at 10.5 Hz with ouabain (Figure 1). Cardiac variables were calculated either manually or achieved using the LabChart blood pressure module (MLS370/8, ADInstruments), and they were averaged over the 10 s period. Left ventricular developed pressure was calculated as the difference between average maximum balloon pressure and end-diastolic pressure, and it was used to measure left ventricular function [35]. To estimate both the cardiac work normalized to heart rate and the cardiac work related to one heart cycle, the rate–pressure product and the systolic pressure-time index was calculated [36,37,38]. The rate–pressure product was calculated as the product of left ventricular developed pressure and heart rate [36]. The systolic pressure–time index was defined as the average pressure during systole multiplied by the systolic duration [37,39]. The maximum first derivative of the left ventricular pressure dP/dtmax was used to estimate left ventricular contractility [40,41]. Similarly, the minimum first derivative of the left ventricular pressure dP/dtmin was used to estimate the efficiency of left ventricular relaxation [42]. Glanz’s calculation for tau (τ_Glanz_) describes pressure decay during isovolumetric relaxation and, accordingly, it was used to gauge the efficiency of the isovolumetric relaxation [42].

### 2.4. Stereology

Stereological methods and principles were used to evaluate morphological changes of left ventricular tissue [43,44,45]. Six WT and five α_2_^+/G301R^ hearts were fixed with 4% paraformaldehyde in the anesthetized mice by perfusion–fixation. In brief, mice were anesthetized with intraperitoneal pentobarbital injection (140 mg/kg; Virbac, Kolding, Denmark). Thoracotomy was performed on the deeply anesthetized mice while their hearts were still beating. A needle connected to a reservoir bag with phosphate buffered saline (PBS; Appendix A) was inserted in the left ventricle through the apex of the heart. The heart and the vascular system were then flushed with PBS under hydrostatic pressure of approximately 100 cm H_2_O. The right atrium was cut open to enable the outflow of the perfusate. When the effluent was clear of erythrocytes, the PBS was swapped to 4% paraformaldehyde and flushing continued. Proper fixation was evident by the paling and stiffening of tissues. The hearts were then dissected and stored in 4% paraformaldehyde for one day before transfer to PBS, until embedding.

The volume of left ventricular tissue was estimated by dividing the left ventricle weight by the density mass of muscle tissue—1.06 g/cm^3^ [46]. The fixed left ventricles were serially sectioned into 2-mm-thick slices, which were subsequently sectioned at a 90° angle into 2 × 2 mm^2^ tissue blocks. The small tissue blocks of the left ventricle were sampled by a smooth fractionator design, sliced to 2.5 µm thick paraffin sections, and then mounted on Superfrost plus slides (Menzel-Gläser, Braunschweig, Germany). Myocyte cell borders were labeled with monoclonal mouse anti-pan cadherin antibody (1:1000; Sigma–Aldrich, Burlington, MA, USA, Merck Life Science A/S, Søborg, Denmark; in PBS with 1% bovine serum albumin (BSA), 0.15% Triton X-100 (Sigma–Aldrich), and wheat germ agglutinin (1:100; Sigma–Aldrich). After washing, the sections were incubated with a mix of horseradish peroxidase-conjugated secondary antibodies (1:200; DAKO A/S, Glostrup, Denmark). The peroxidase staining was visualized by 0.05% 3,3′-diaminobenzidine tetrahydrochloride dissolved in PBS with 0.1% H_2_O_2_. Nuclei were visualized by counterstaining with Mayer’s hematoxylin.

Unbiased counting frames were systematically, uniformly, and randomly superimposed onto the live images of the tissue sections with the newCAST software (Visiopharm A/S, Hørsholm, Denmark) using a light microscope with a motorized stage (BX51 microscope; DP70 camera; UPlanFI (NA 0.50) ×20 dry lens, Olympus, Tokyo, Japan). Cardiomyocyte profiles sampled by the unbiased counting frames were counted. Cardiomyocytes were assumed to be isotropic on a global scale, and their volumes were estimated with the 3D nucleator. No correction for tissue shrinkage was made. To estimate fibrosis, sections were stained with Masson’s trichrome and analyzed using test points (Visiopharm), as described previously [43]. In brief, two-point grids—(i) a 12 × 12 point grid for fibrosis points count (P_fib_) and (ii) a 3 × 3 point grid for the left ventricle volume points count (P_LV_)—were used. The volume fraction of fibrosis was calculated for each animal as (∑P_fib_/∑P_LV_)•(9/144) [43]. The total fibrosis volume was calculated by multiplying the volume fraction of fibrosis with the volume of left ventricle tissue [43]. The area fraction of all blood vessels was estimated by measuring the averaged short axis lumen area per region of interest.

### 2.5. Statistical Analysis

All data are presented as the mean value ± standard error of the mean (SEM). All data that violated the assumption of normality were analyzed with a non-parametric test. Stereology data and ex vivo cardiovascular variables at baseline and during atrial pacing were compared with an unpaired *t*-test or a Mann–Whitney test. Time-control experiments were analyzed with a paired *t*-test or a Wilcoxon test. Cardiac variables in the absence of and in the presence of ouabain were analyzed with a multiple Wilcoxon test or a two-way repeated-measures ANOVA, and Sidaks’s multiple comparisons tests were used when appropriate if a statistically significant effect of a variable was detected. *p*-values below 0.05 were considered statistically significant, and *n* indicates the number of animals per dataset.

## 3. Results

### 3.1. Isolated Hearts from α_2_^+/G301R^ Mice Displayed Greater Contractility Than WT Hearts

Isolated hearts from WT and α_2_^+/G301R^ mice were studied in the Langendorff perfusion, and the cardiac variables of the two genotypes were compared during sinus rhythm, atrial pacing at 10.5 Hz (630 BPM), and in the presence of ouabain (Figure 2).

The set end-diastolic pressure (WT; 8.2 ± 0.3 mmHg vs. α_2_^+/G301R^; 8.8 ± 0.5 mmHg; *p* = 0.2939) of the WT and α_2_^+/G301R^ hearts was not different. At baseline, there was no difference between the two genotypes in the sinus rate (Figure 3a).

During sinus rhythm, the α_2_^+/G301R^ hearts exhibited greater left-ventricular-developed pressure (*p* = 0.0101), higher left ventricular dP/dt_max_ (*p* = 0.0332), and lower left ventricular dP/dt_min_ (*p* = 0.0350) than WT hearts (Figure 3b–d). The rate pressure product was not different between the two groups; however, the systolic pressure–time index was higher (*p* = 0.0023) in α_2_^+/G301R^ hearts than in WT hearts (Figure 3e,f). There was no difference in τ_Glanz_ between the two groups, but the systolic duration in α_2_^+/G301R^ hearts (*p* = 0.0184) was longer than the durations in the WT hearts (Figure 3g,h).

When the hearts were paced at 10.5 Hz, all differences between α_2_^+/G301R^ and WT variables, including left-ventricular-developed pressure, left ventricular dP/dt_max_, left ventricular dP/dt_min_, the systolic pressure–time index, and the systolic duration were eliminated, suggesting rate-dependency (Figure 4).

### 3.2. Ex Vivo Cardiac Function Was Maintained in the Presence of Ouabain during Sinus Rhythm in Both WT and α_2_^+/G301R^ Hearts

Ouabain (1 µM) was added to the perfusate, and 10 min after, changes to cardiac variables of α_2_^+/G301R^ and WT hearts were compared during sinus rhythm. The heart rate did not change in the presence of ouabain for either WT or α_2_^+/G301R^ hearts (Figure 5a). In the presence of ouabain, the left-ventricular-developed pressure of α_2_^+/G301R^ hearts remained higher (*p* = 0.0274) than that of WT hearts (Figure 5b). During the period after atrial pacing, the difference in left ventricular dP/dt_max_ between the genotypes did not achieve statistical significance (*p* = 0.0652 in the absence of ouabain and *p* = 0.0513 in the presence of ouabain; Figure 5c). The left ventricular dP/dt_min_ remained more negative in α_2_^+/G301R^ hearts than in WT hearts in the presence of ouabain (*p* = 0.0200; Figure 5d). The rate–pressure product of α_2_^+/G301R^ hearts became significantly higher than that of WT hearts in the presence of ouabain (*p* = 0.0231; Figure 5e). Accordingly, the systolic pressure–time index was increased in α_2_^+/G301R^ hearts in the presence of ouabain (*p* = 0.0365), and it remained higher in α_2_^+/G301R^ hearts than in WT hearts, both in the absence of and the presence of ouabain (*p* = 0.0181; Figure 5f). Neither τ_Glanz_ nor the systolic duration changed in the presence of ouabain (Figure 5g,h).

In the time-control experiments, heart rate was not changed, but all functional cardiac variables, including left-ventricular-developed pressure (*p* = 0.0040), left ventricular dP/dt_max_ (*p* = 0.0085), left ventricular dP/dt_min_ (*p* = 0.0072), rate–pressure product (*p* = 0.0005), and systolic pressure–time index (*p* = 0.0038) deteriorated over time with exposure to pacing (Appendix A). No changes were observed in τ_Glanz_ and systolic duration over time in the time-control experiments (Appendix A).

### 3.3. Contractility during Atrial Pacing Was Augmented in α_2_^+/G301R^ Hearts in the Presence of Ouabain, and This Was Associated with Increased Systolic Work

During atrial pacing at 10.5 Hz (630 BPM) and in the presence of ouabain, the left-ventricular-developed pressure (*p* = 0.0319) and the left ventricular dP/dtmax (*p* = 0.0313) of the α_2_^+/G301R^ hearts were larger than those of WT hearts (Figure 6a,b). There was no difference in left ventricular dP/dt_min_ between the two groups (Figure 6c). The rate–pressure product was larger in α_2_^+/G301R^ hearts than in WT hearts during atrial pacing in the presence of ouabain (*p* = 0.0319), but the difference in the systolic pressure-time index did not achieve statistical significance (*p* = 0.0577; Figure 6d,e). During atrial pacing and in the presence of ouabain, τ_Glanz_ and systolic duration were not different between the two genotypes (Figure 6f,g).

In the time-control experiments, the left-ventricular-developed pressure (*p* = 0.0306) and left ventricular dP/dt_max_ (*p* = 0.0436) declined between the first and second atrial pacing sessions (Appendix A). The left ventricular dP/dt_min_ remained unchanged (Appendix A). The rate–pressure product (*p* = 0.0305) and the systolic pressure–time index (*p* = 0.0077) also deteriorated between the first and second atrial pacing sessions (Appendix A). Between the first and the second atrial pacing sessions, no change was observed in τ_Glanz_, but the systolic duration was slightly shortened (*p* = 0.0182) in the time-control experiments (Appendix A).

### 3.4. Contractile Cardiac Phenotype of α_2_^+/G301R^ Mice Was Not Due to Morphological Changes in the Heart

The stereological assessment of dissected hearts from WT and α_2_^+/G301R^ mice revealed no differences in volume of the left ventricular wall (Figure 7a–c). Furthermore, the number of cardiomyocytes normalized to area and their volume in the two groups were similar (Figure 7d,e). There were no differences between the WT and α_2_^+/G301R^ hearts in the amount of fibrosis or the degree of vascularization of the left ventricle (Figure 7f–h).

## 4. Discussion

In the current study, we investigated the Na^+^,K^+^-ATPase α_2_-isoform-specific contribution to the regulation of cardiac contractility in the context of the FHM2-associated mutation, α_2_^+/G301R^. We used aged (>8-month-old) α_2_^+/G301R^ mice and compared them with matching WT. Previously, we described the hypercontractile vasculature and dilation of the left ventricle associated with a reduction in the ejection fraction in aged α_2_^+/G301R^ mice [26,27,47]. However, these studies were either conducted on isolated vascular tissues or carried out in an in vivo setting in anesthetized mice. Thus, a specific assessment of the isolated cardiac function of α_2_^+/G301R^ mice was lacking.

The previously reported inverse expression of the α_1_- and α_2_-isoforms in the α_2_^+/G301R^ hearts may be helpful in identifying the importance of each isoform for cardiac contractility. This inverse proportionality between α_1_- and α_2_-isoform expressions was also reported in other mouse models with genetic manipulation of the Na^+^,K^+^-ATPase [48]. Importantly, in other mouse models with genetic manipulation of the α_1_- and/or α_2_-isoform expression, the expression of proteins associated with Ca^2+^ cycling in cardiomyocytes was also changed [48,49]. The change in expression of Ca^2+^ cycling proteins, e.g., NCX and sarco/endoplasmic reticulum Ca^2+^-ATPase (SERCA), may in itself affect cardiomyocyte contractility and, therefore, is important to consider [50]. According to Western blot and proteomics data, we previously showed that the expression of the NCX and SERCA in the hearts of WT mice and α_2_^+/G301R^ mice were similar [26]. On this basis, we propound that the opposing change in α_1_- and α_2_-isoform expressions in α_2_^+/G301R^ mice may yield two hypothetical scenarios that may illuminate the preferential isoform regulating cardiac contractility: (i) if the α_1_-isoform is important in defining cardiac contractility, its increased expression will decrease the intracellular Na^+^ concentration, leading to increased NCX-mediated Ca^2+^-extrusion and reduced cardiac contractility; or (ii) if the α_2_-isoform defines cardiac contractility, the reduced expression of the α_2_-isoform leads to an increase in the intracellular Na^+^ concentration, with a subsequent decrease in NCX-mediated Ca^2+^-extrusion and, thus, an increase in cardiac contractility.

Here, we found that during sinus rhythm, isolated hearts from α_2_^+/G301R^ mice developed a greater left ventricular pressure and left ventricular dP/dt_max_ than those of WT hearts, indicating increased contractility of α_2_^+/G301R^ hearts. This is in accordance with the hypothetical scenario (ii) above, which suggests that the α_2_-isoform is the preferred isoform for cardiac contractility regulation. This is in line with previous research demonstrating that heterozygous knockout of the Na^+^,K^+^-ATPase α_2_-isoform is associated with increased cardiac contractility in mice [51].

Greater cardiac contractility is associated with higher myocardial oxygen consumption [52]. Although not a direct measure, cardiac work and, thus, myocardial oxygen consumption may be gauged by the systolic pressure–time index [37,39,53]. The systolic pressure–time index was notably higher in the hearts from α_2_^+/G301R^ mice, suggesting a larger myocardial oxygen consumption. This indication is in line with the previously reported larger myocardial oxygen consumption in cardiac tissue from aged α_2_^+/G301R^ mice, which was based on mitochondrial respirometry analysis [26]. When considering the cardiac pressure work normalized to heart rate, however, there was no significant difference between the two genotypes alluding to a rate-dependent cardiac change. This was further suggested by the longer systolic durations of α_2_^+/G301R^ hearts than those of WT hearts. Based on this, we tested whether the greater cardiac contractility of α_2_^+/G301R^ hearts was rate-dependent by atrial pacing. Strikingly, all cardiac variables of the two genotypes were equalized during atrial pacing, further confirming the rate-dependent cardiac changes in α_2_^+/G301R^ hearts. We suggest that the heart rate may accommodate and offset the overall energy cost of the greater contractility in a compensatory manner in α_2_^+/G301R^ hearts, compared with WT hearts.

We previously reported a lower ejection fraction in aged α_2_^+/G301R^ mice in vivo than in matching WT mice [26]. However, in an in vivo setting, vascular, humoral, and neuronal factors must also be considered in conjunction with cardiac function [54,55]. Since the ejection fraction is calculated based on ventricular volumes, dilation of the ventricle may skew this index without any significant change in contractility [56]. Hence, the reduced ejection fraction may reflect either an elevated afterload, the dilation of the ventricles previously described for α_2_^+/G301R^ mice, or a combination of the two [26,27,47]. Thus, we suggest that the greater cardiac contractility in α_2_^+/G301R^ mice is masked by peripheral factors in vivo under resting conditions.

In our Langendorff protocol, 1 µM of ouabain was added to the perfusate to further investigate the isolated contribution of the α_2_-isoform to cardiac function [57]. Considering the deterioration of functional cardiac variables in the time-control experiments, we suggest that the maintenance of cardiac function in the presence of ouabain may indicate its positive inotropic effects [58]. Interestingly, ouabain increased the systolic work only in the α_2_^+/G301R^ hearts. In addition, during atrial pacing in the absence of ouabain, all cardiac variables between the two genotypes were similar, but in the presence of ouabain, cardiac contractility was more enhanced in α_2_^+/G301R^ hearts than in WT hearts during atrial pacing. We suggest that this may indicate an augmented inotropic effect of ouabain in α_2_^+/G301R^ hearts. These results may also suggest the rate-independence of the acute inotropic effects of ouabain on isolated hearts.

We previously proposed that the augmented ouabain response in vivo in α_2_^+/G301R^ mice is due to decreased expression of the Na^+^,K^+^-ATPase α_2_-isoform [24,25,26,27]. This will result in a larger fractional occupancy of the α_2_-isoform at a given ouabain concentration and, thereby, increase the effect of ouabain [28,29]. We suggest that this may also explain a more augmented contractility of α_2_^+/G301R^ hearts than that of WT hearts during atrial pacing in the presence of ouabain.

The presence of ouabain at a concentration specific for acute α_2_-isoform inhibition did not change the temporal cardiac variables, including heart rate and systolic duration. This suggests that the mechanisms underlying the greater contractility of α_2_^+/G301R^ hearts under resting conditions and the inotropy of ouabain are different; they may reflect chronic adaptive changes and the acute effect of reduced α_2_-isoform function, respectively. Previous evidence indicates that the cardiometabolic profile was changed in genetically modified mouse models under conditions of chronic elevation of intracellular Na^+^ concentration ([Na^+^]_in_), which was associated with altered mitochondrial Ca^2+^ handling [59]. In addition, this metabolic change involved the switch from fatty acid oxidation to glycolysis as the primary means of energy utilization, which was also reported for aged α_2_^+/G301R^ mice, perhaps suggesting a similar mechanism at play [26]. In contrast, in the previous report, acute elevation of [Na^+^]_in_ by ouabain (75 µM) did not change the cardiometabolic profile, supporting the notion of distinct mechanisms of chronic and acute elevations of [Na^+^]_in_ [59]. Although the bulk cytosolic [Na^+^]_in_ has been proposed to be primarily regulated by the α_1_-isoform, our study suggests that the regulation of [Na^+^]_in_ associated with the α_2_-isoform is significant for the cardiometabolic and functional profile of α_2_^+/G301R^ hearts [26,27].

We propose two possible underlying mechanisms that may involve the Na^+^,K^+^-ATPase α_2_-isoform. First, the acute effect of ouabain may be effectuated via the conventional Na^+^,K^+^-ATPase/NCX pathway modulating [Ca^2+^]_in_ [8,13]. Similarly, the greater baseline contractility in α_2_^+/G301R^ hearts during sinus rhythm, compared with WT hearts, may also be explained as a mechanism of this pathway. Specifically, a reduced abundance of the α_2_-isoform may diminish the NCX activity with consequent elevation of [Ca^2+^]_in_. However, previously, no difference in membrane conductance between α_2_^+/G301R^ and WT cardiomyocytes was found [26]. In conjunction with the protein expression analysis indicating no changes in other ion-transporting proteins, e.g., the NCX, this may suggest that a significant change in intracellular ion homeostasis is unlikely [26]. Thus, the mechanism underlying the greater contractility of α_2_^+/G301R^ hearts at baseline during sinus rhythm may be mediated by another mechanism, e.g., the Na^+^,K^+^-ATPase-associated intracellular signaling [60,61]. Elaboratively, the inhibition of the Na^+^,K^+^-ATPase α_2_-isoform with low-dose ouabain (10 μM) has previously been shown to activate the Src kinase, a non-receptor tyrosine kinase [62]. Src activation has been associated with the activation of the ERK1/2 pathway [61,63]. In cardiomyocytes, the activation of this pathway has been associated with increased contractility due to changed Ca^2+^ handling [61]. In a previous study, we demonstrated that hearts from aged α_2_^+/G301R^ mice (>8 months old) display an increased capacity for Src activation together with an increased expression of Src and ERK1/2 kinases, compared with hearts from matching WT mice [26]. An increased capacity for Src activation may lead to cardiac alterations resulting in greater cardiac contractility [61]. The elucidation of these distinct mechanisms underlying the acute enhancement of contractility in α_2_^+/G301R^ hearts in the presence of ouabain and their larger cardiac contractility at baseline, compared with WT hearts, remains an important objective for future research.

The contraction and relaxation phases of the heart are intimately connected [64]. Imbalances in this relationship may result in cardiac pathology [65,66]. For instance, if the rate of myocardial relaxation is not matched with the rate of contraction, this may lead to diastolic dysfunction [65,67]. Therefore, we also assessed diastolic function, i.e., left ventricular dP/dt_min_ and τ_Glanz_. In rodents, there is a tight coupling between the left ventricular dP/dt_max_ and left ventricular dP/dt_min_ [68]. Accordingly, we found that the higher left ventricular dP/dt_max_ of α_2_^+/G301R^ hearts was matched with a lower left ventricular dP/dt_min_ when compared with the values of WT hearts. However, despite a more negative left ventricular dP/dt_min_ for α_2_^+/G301R^ hearts in comparison with WT, this did not translate into a lower τ_Glanz_ value. Notably, the left ventricular dP/dt_min_ only describes the fastest deceleration rate of pressure for a single time-point [42], whereas τ_Glanz_ provides a general description of the relaxation rate [42]. Therefore, we suggest that the diastolic function is not significantly changed in α_2_^+/G301R^ hearts and that it does not mirror the changes in their augmented cardiac contractility.

The diastolic duration, the vascular density, and the pressure gradient between aortic and ventricular pressures during diastole are critical determinants of coronary perfusion under physiological settings [69]. Surprisingly, despite the greater systolic work of α_2_^+/G301R^ hearts, their diastolic durations, vascular densities, and τ_Glanz_ values were not different when compared with those observed in WT mice. The coronary function of the two genotypes was not assessed in the current study. The prefusion pressure was set to 60 mmHg and maintained throughout the experiment. We cannot exclude differences in coronary flow; however, if the function of the coronary circulation is similar between the two genotypes, the diastolic function may not be sufficiently promoted to compensate for the larger systolic myocardial oxygen consumption in the α_2_^+/G301R^ hearts.

Several conditions may restrict diastolic function, e.g., myocardial scarring, fibrosis, or mitochondrial dysfunction [70]. In our study, the fibrosis density volume was similar between the two groups. Thus, it is not likely that fibrosis accounts for a potential restriction of diastolic function in α_2_^+/G301R^ hearts. Intriguingly, mitochondrial dysfunction was previously posited as part of the cardiac phenotype of aged α_2_^+/G301R^ mice [26]. Notably, sustained cardiac workload has been associated with mitochondrial dysfunction and the increased generation of reactive oxygen species in the cardiac tissue, conditions also described for aged α_2_^+/G301R^ mice in vivo [26,27,71,72]. Therefore, we suggest that the altered cardiac work during systole underlies the cardiometabolic phenotype of aged α_2_^+/G301R^ mice [26]. Whether the mitochondrial dysfunction may underlie potential diastolic restriction in aged α_2_^+/G301R^ mice remains an important objective for future research.

An increasing number of reports associate mutations in the Na^+^,K^+^-ATPase with pathological conditions in humans [73,74]. Thus, the investigation of isoform-specific contributions of the Na^+^,K^+^-ATPase to human pathology and physiology is pertinent, and it will allow for the development of an improved and targeted armamentarium against Na^+^,K^+^-ATPase-associated pathologies [8,73,74]. In this study, we described the significant role of α_2_-isoform function for cardiac contractility. Notably, this study does not dismiss the inotropic property of the functional inhibition of the α_1_-isoform; rather, this study suggests a significant role of the α_2_-isoform in regulating cardiac contractility, in accordance with existing literature [8,13]. However, the current work is distinguished from previous studies by describing the importance of the α_2_-isoform for cardiac contractility and efficiency in the context of FHM2, i.e., a clinically relevant pathology in humans [75].

Previously, we also demonstrated the hypercontractile vasculature of α_2_^+/G301R^ mice and demonstrated this to be associated with the α_2_-isoform function in vivo [27,47]. We proposed that the FHM2-assocaited contractile cardiac and vascular phenotypes in α_2_^+/G301R^ mice underlie their cardiometabolic derangements [26,27]. Whether FHM2 migraineurs exhibit a similar cardiometabolic and vascular phenotype as α_2_^+/G301R^ mice remains to be investigated. Many of the cardiac changes described in the α_2_^+/G301R^ mice are subtle and may develop slowly over time, similar to diastolic heart failure with no obvious clinical symptoms [26,27,66,76,77]. Thus, if similar cardiac changes are indeed implicated in FHM2 migraineurs, this calls for heightened clinical vigilance and investigation of cardiac health.

## 5. Conclusions

In conclusion, the isolated hearts of α_2_^+/G301R^ mice displayed a rate-dependent hypercontractile cardiac phenotype compared with the hearts of WT mice. In the presence of ouabain (1 µM), systolic work was only increased in α_2_^+/G301R^ hearts. During atrial pacing, α_2_^+/G301R^ hearts displayed more augmented contractility than WT hearts in the presence of ouabain. This indicated the significant role of the α_2_-isoform in the regulation of cardiac contractility in the context of FHM2. We propose that the greater systolic work of α_2_^+/G301R^ hearts is associated with an increased myocardial oxygen demand, which over time may underlie and lead to their deranged cardiometabolic phenotype.

## Figures and Tables

**Figure 1 cells-12-01108-f001:**
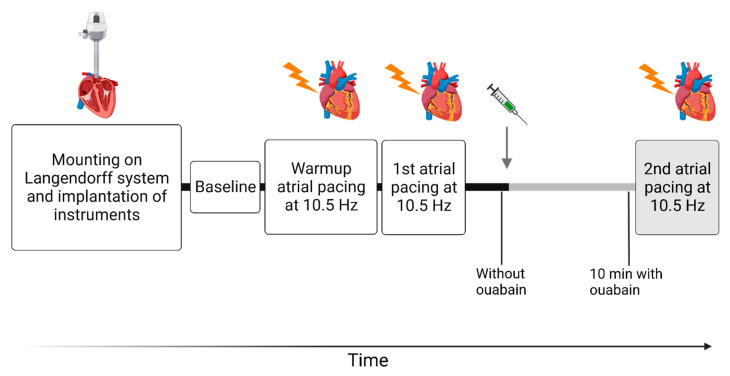
**Overview of the protocol assessing the isolated cardiac function.** The gray arrow indicates the addition of ouabain to the perfusate (1 µM) and the gray bars indicate the subsequent period in the presence of ouabain. First, hearts were excised and mounted in the Langendorff perfusion for functional assessment. Once stabilized, baseline recordings were measured. After the “warm-up” pacing, the protocol contained two sessions of atrial pacing at 10.5 Hz (630 BPM). The first atrial pacing was carried out in the absence of ouabain, and the second pacing session was carried out in the presence of ouabain. These two pacing sessions were carried out in the absence of ouabain in the time-control experiments. In between the first and second atrial pacing sessions, cardiac variables were measured during sinus rhythm in the absence of ouabain and 10 min after ouabain was added to the perfusate. At similar time points, cardiac variables were measured in the time-control experiments, but without ouabain. Figure 1 was created with BioRender.com.

**Figure 2 cells-12-01108-f002:**
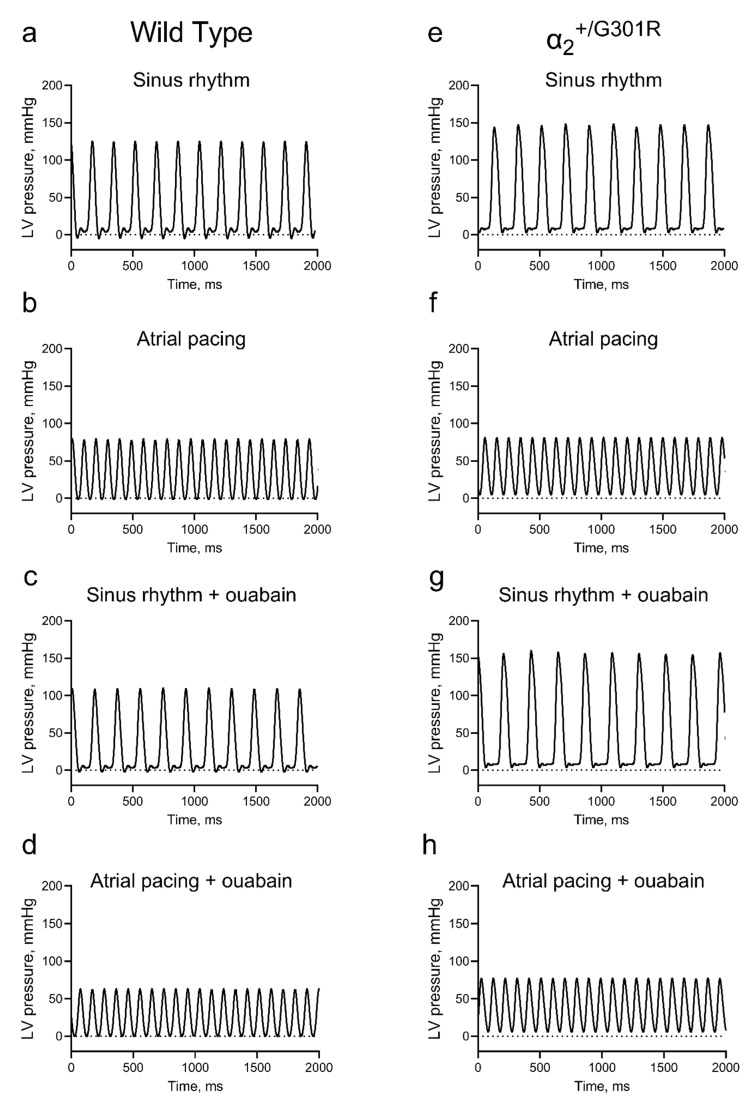
**Representative traces of left-ventricular-pressure development of the isolated α_2_^+/G301R^ and wild-type (WT) hearts.** The left ventricular (LV) pressure waveforms of isolated hearts from WT mice were assessed during (**a**) sinus rhythm in the absence of ouabain, (**b**) pacing at 10.5 Hz (630 BPM), (**c**) sinus rhythm in the presence of ouabain, and (**d**) pacing at 10.5 Hz in the presence of ouabain. Similarly, the LV pressure waveforms of isolated hearts from α_2_^+/G301R^ mice were assessed during (**e**) sinus rhythm, (**f**) pacing at 10.5 Hz, (**g**) sinus rhythm in the presence of ouabain, and (**h**) pacing at 10.5 Hz in the presence of ouabain.

**Figure 3 cells-12-01108-f003:**
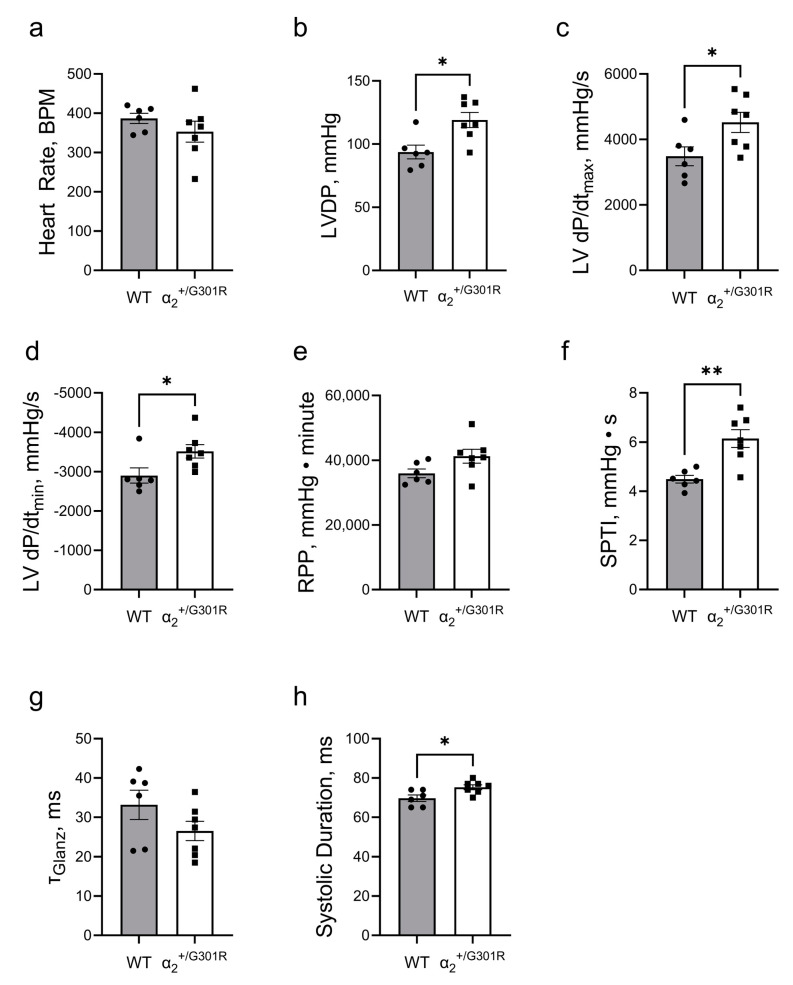
**Isolated hearts from α_2_^+/G301R^ mice exhibited greater cardiac contractility and increased systolic work during sinus rhythm**. (**a**) Heart rate was similar between WT and α_2_^+/G301R^ hearts. (**b**) Left-ventricular-developed pressure (LVDP) and (**c**) left ventricular dP/dt_max_ (LV dP/dt_max_) were greater for α_2_^+/G301R^ hearts than for WT hearts. (**d**) Left ventricular dP/dt_min_ (LV dP/dt_min_) was more negative for α_2_^+/G301R^ hearts than for WT hearts. (**e**) The rate–pressure product (RPP) was not different between the two genotypes, but (**f**) the systolic pressure–time index (SPTI) was larger in α_2_^+/G301R^ hearts than in WT hearts. (**g**) τ_Glanz_ was not different between the two groups, but (**h**) the systolic duration was longer for α_2_^+/G301R^ hearts. Data were compared with an unpaired *t*-test or a Mann–Whitney test. * *p* < 0.05 and ** *p* < 0.01; wild type vs. α_2_^+/G301R^. *n* = 6–7.

**Figure 4 cells-12-01108-f004:**
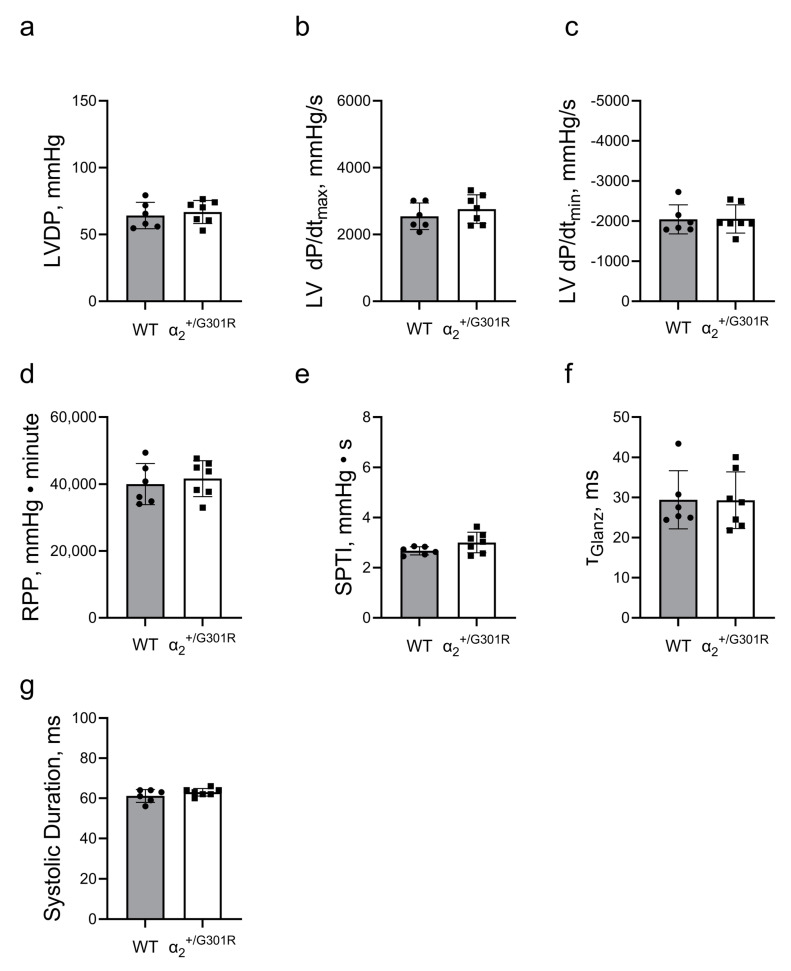
**Differences in cardiac contractility between α_2_^+/G301R^ hearts and WT hearts were eliminated during atrial pacing at 10.5 Hz (630 BPM).** Cardiac variables including (**a**) left-ventricular-developed pressure (LVDP), (**b**) left ventricular dP/dt_max_ (LV dP/dt_max_), (**c**) left ventricular dP/dt_min_ (LV dP/dt_min_), (**d**) rate–pressure product (RPP), (**e**) systolic pressure–time index (SPTI), (**f**) τ_Glanz_, and (**g**) systolic duration were not different between α_2_^+/G301R^ hearts and WT during atrial pacing. Data were compared with an unpaired *t*-test or a Mann–Whitney test. *n* = 6–7.

**Figure 5 cells-12-01108-f005:**
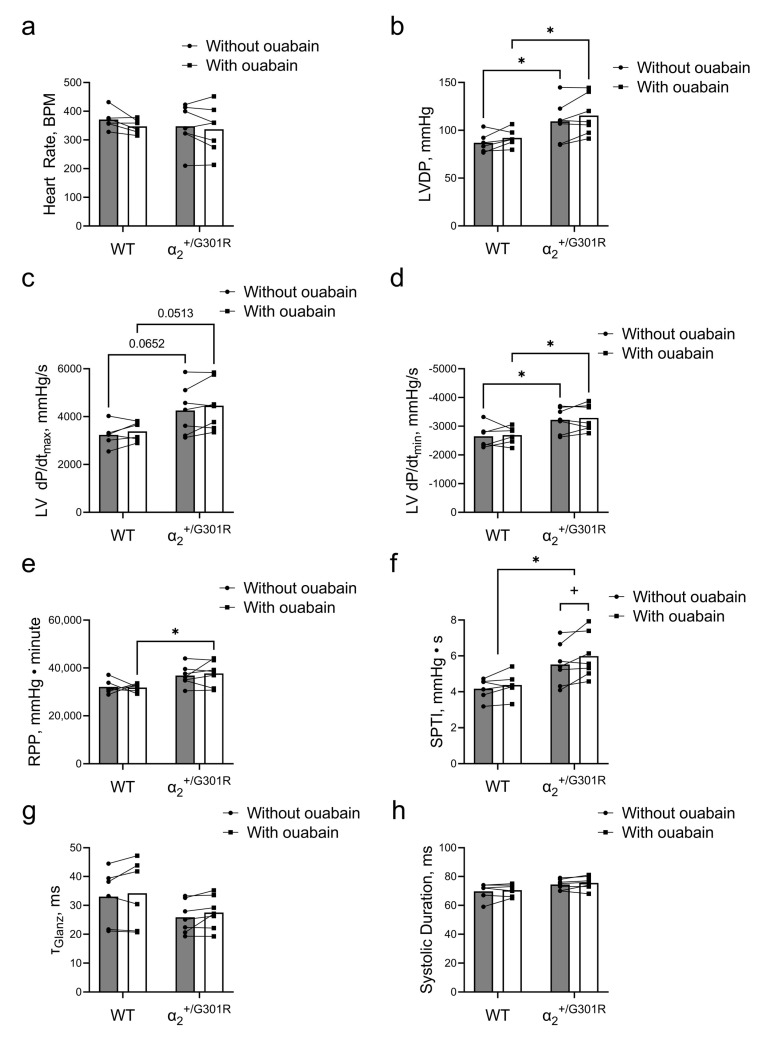
**In the presence of ouabain during sinus rhythm, systolic work increased only for α_2_^+/G301R^ hearts and not for WT hearts**. (**a**) Heart rate was unaffected in the presence of ouabain in both genotypes. (**b**) The left-ventricular-developed pressure (LVDP) of α_2_^+/G301R^ hearts remained greater than that of WT hearts. (**c**) The differences in left ventricular dP/dt_max_ (LV dP/dt_max_) between the genotypes did not achieve statistical significance in the absence of (*p* = 0.0652) or in the presence of ouabain (*p* = 0.0513). (**d**) The difference in left ventricular dP/dt_min_ (LV dP/dt_min_) between WT and α_2_^+/G301R^ hearts remained in the presence of ouabain. (**e**) The rate–pressure product (RPP) between the two genotypes was different only in the presence of ouabain, and (**f**) the systolic pressure–time index (SPTI) remained larger for α_2_^+/G301R^ hearts than for WT hearts and it was only increased for α_2_^+/G301R^ hearts in the presence of ouabain. (**g**) τ_Glanz_ and (**h**) the systolic duration did not change in the presence of ouabain. Data were compared with a two-way repeated-measures ANOVA followed by Sidak’s multiple comparison test. * *p* < 0.05; WT vs. α_2_^+/G301R^ variables and ^+^
*p* < 0.05; comparison of intragroup variables. *n* = 6–7.

**Figure 6 cells-12-01108-f006:**
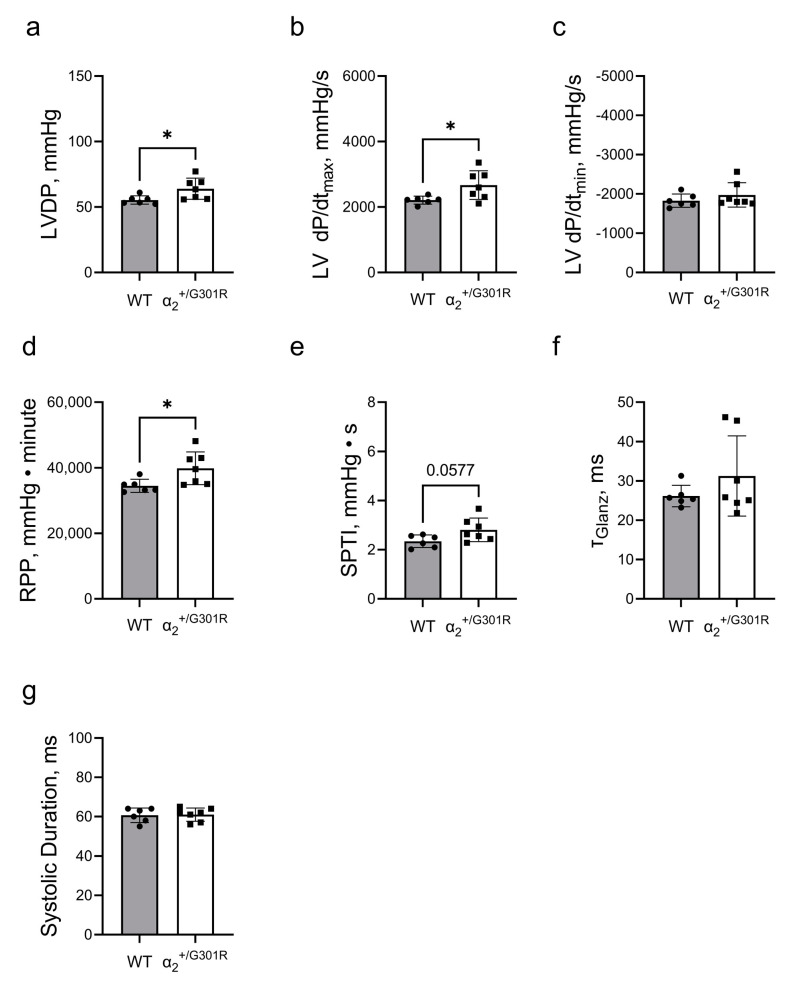
**In the presence of ouabain cardiac contractility was more enhanced in α_2_^+/G301R^ hearts than in WT hearts during atrial pacing at 10.5 Hz (630 BPM).** (**a**) Left-ventricular-developed pressure (LVDP) and (**b**) the left ventricular dP/dt_max_ (LV dP/dt_max_) of α_2_^+/G301R^ hearts were greater than those of WT hearts during atrial pacing in the presence of ouabain. (**c**) There was no difference between the groups in left ventricular dP/dt_min_ (LV dP/dt_min_). (**d**) The rate–pressure product (RPP) was larger for α_2_^+/G301R^ hearts than for WT hearts, but (**e**) the difference in the systolic pressure–time index (SPTI) between the two genotypes did not achieve statistical significance (*p* = 0.0577) during atrial pacing and in the presence of ouabain. There were no differences in (**f**) τ_Glanz_ and (**g**) systolic duration between the two groups. Data were compared with an unpaired *t*-test or a Mann–Whitney test. * *p* < 0.05; WT vs. α_2_^+/G301R^ variables. *n* = 6–7.

**Figure 7 cells-12-01108-f007:**
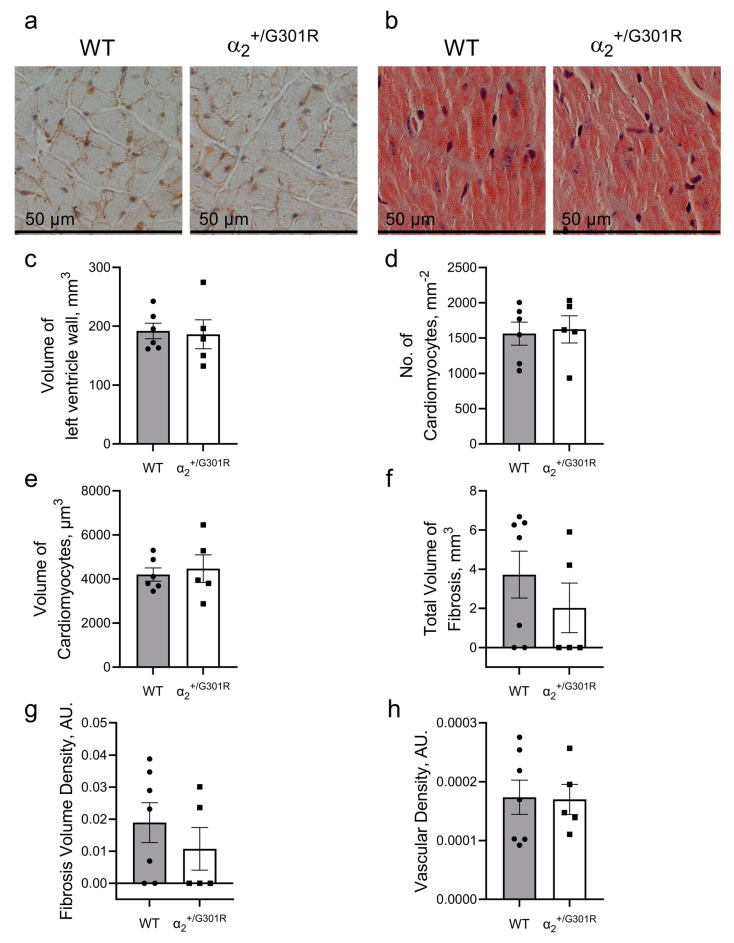
**The hearts of 8-month-old α_2_^+/G301R^ mice are morphologically similar to those of age-matched WT hearts.** The morphology of the left ventricles from aged α_2_^+/G301R^ and WT mice were assessed stereologically. (**a**) Representative image of left ventricular tissue from WT and α_2_^+/G301R^ hearts that were used for cardiomyocyte characterization. (**b**) Representative Masson’s trichrome staining of WT and α_2_^+/G301R^ heart slices for estimation of fibrosis. There were no differences between the two groups in (**c**) volume of left ventricular wall, (**d**) number of cardiomyocyte profiles per area, (**e**) volume of cardiomyocytes, (**f**) total volume of fibrosis, (**g**) density of fibrosis, or (**h**) vascular density of the left ventricular wall. Data were compared with an unpaired *t*-test or a Mann–Whitney test. *n* = 5–7.

## Data Availability

Not applicable.

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
