# Peer review of "Hypercontractile Cardiac Phenotype in Mice with Migraine-Associated Mutation in the Na+,K+-ATPase α2-Isoform"

_cells, 2023, doi:10.3390/cells12081108_

Round 1

Reviewer 1 Report

The manuscript „Hypercontractile Cardiac Phenotype in Mice with Migraine-associated Mutation in the Na+,K+-ATPase α2-isoform“ by Rajkumar Rajanathan, Clàudia Vilaseca i Riera, Tina Myhre Pedersen, Christian Staehr, Elena V. Bouzinova, Jens Randel Nyengaard, Morten B. Thomsen, Hans Erik Bøtker, and Vladimir V. Matchkov presents new data showing that the contractility and the pro-contractile effect of low concentrations of ouabain are enhanced in isolated hearts of α2+/G301R mice.

This referee has the following comments:

  1. Introduction: It is stated that α2+/G301R mice express less α2-subunits and that low ouabain concentrations affect mainly this subunit. It is unclear how this leads to the hypothesis that isolated hearts from α2+/G301R mice are more contractile and more responsive to ouabain than hearts from wild-type (WT) mice. What do you mean with “more responsive to ouabain” when the expression of the α2-isoform is reduced? Please explain this in the introduction and not only in the discussion (p.20, para 4).
  2. Introduction: The statement “This study demonstrates the preferential role of the α2-isoform in the regulation of cardiac contractility in a murine model…” is unclear. There was no comparison with the role of the α1-isoform.
  3. Methods. How did you assess possible effects of ouabain on coronaries, did you observe changes in perfusion pressure?
  4. Methods, statistics: Please describe n. Does it represent biological replicates?
  5. Results, p.8, para1: When comparing the set end-diastolic pressure, may be an equivalence test would have been more appropriate?
  6. Results, p.8, para3: It makes no sense to have a similar cycle duration with longer systolic duration and non-different diastolic duration as systolic duration + diastolic duration = cycle length. Given the variability in diastolic duration and cycle length, did you perform a sample size calculation for these parameters?
  7. Results, Fig. 7c: In the legend it is stated that “The difference between left ventricular dP/dtmax (LVdP/dtmax) were similar between the genotypes.” but this difference is not shown. Is there a significant difference between the genotypes in the absence and in the presence of ouabain as shown in Fig. 3?
  8. Results, Fig. 7f: The statement “the systolic pressure-time index (SPTI) remained larger for α2+/G301R hearts compared to WT hearts” is unclear as no significant difference is indicated in contrast to Fig. 3.
  9. Results, p.12, para1: Here you state, for example: “After atrial pacing, the difference in left ventricular dP/dtmax be-303 tween genotypes did not achieve statistical significance during this period (P = 0.0652 without ouabain and P = 0.0513 in the presence of ouabain; Figure 7c).” It seems confusing the use statistics with different rigor when comparing data shown in Fig. 3-6 and in Fig. 7-10. Did you perform sample size calculations for the data shown in Fig. 7-10, where probably higher n numbers are required to make statistical inference based on the same effect size as shown in Fig. 3-7?
  10. Results, Fig.S1, S3, S4: Since many parameters changes during time control, the evaluation of the effect of ouabain requires the calculation of the differences (control - ouabain) and (control - time control) and the statistical comparison of these differences. What was seen in time control experiments in mutant mice?
  11. Results, Fig. 9 and 10: Why do you show and compare here only 2 groups, in contrast to the data shown in Fig. 7 and 9? This should be unified.
  12. Results, p.15, para 1: The statement “During atrial pacing at 10.5 Hz (630 BPM), ouabain significantly enhanced the left ventricular developed pressure (P = 0.0319) and the left ventricular dP/dtmax (P = 0.0313) of the α2+/G301R hearts compared to WT hearts” is misleading as there is no comparison between parameters in the absence and presence of ouabain. Please revise. If such a comparison is desired to evaluate the effect of ouabain, the effect of ouabain on the desired parameter should be calculated separately in WT and also in mutant mice and these values compared statistically.

Author Response

We thank the reviewer for the excellent comments and suggestions. We believe they have been helpful in improving the manuscript. Please, find the response to each comment point by point below or in the attached document.

Comments and Suggestions for Authors

The manuscript „Hypercontractile Cardiac Phenotype in Mice with Migraine-associated Mutation in the Na+,K+-ATPase α2-isoform“ by Rajkumar Rajanathan, Clàudia Vilaseca i Riera, Tina Myhre Pedersen, Christian Staehr, Elena V. Bouzinova, Jens Randel Nyengaard, Morten B. Thomsen, Hans Erik Bøtker, and Vladimir V. Matchkov presents new data showing enhanced contractility and potentiated pro-contractile effect of low ouabain concentrations in isolated hearts of α2+/G301R mice.

 This referee has the following comments: 

  • Introduction: It is stated that α2+/G301R mice express less α2-subunits and that low ouabain concentrations affect mainly this subunit. It is unclear how this leads to the hypothesis that isolated hearts from α2+/G301R mice are more contractile and more responsive to ouabain than hearts from wild-type (WT) mice. What do you mean with “more responsive to ouabain” when the expression of the α2-isoform is reduced? Please explain this in the introduction and not only in the discussion (p.20, para 4).

This has now been clarified in the introduction section. Specifically, we have now made it clear why we expect increased contractility of α2+/G301R hearts (lines 71-74 in the revised manuscript), and we have also added an explanation of the hypothesis based on a ‘higher functional occupancy’ for ouabain in α2+/G301R hearts (lines 101-105 in the revised manuscript). Additionally, the hypothesis has been re-phrased to be more specific (lines 110-112 in the revised manuscript).

  • Introduction: The statement “This study demonstrates the preferential role of the α2-isoform in the regulation of cardiac contractility in a murine model…” is unclear. There was no comparison with the role of the α1-isoform.

In this mouse model, the cardiac α1-isoform expression is increased, and the α2-isoform expression is reduced. Thus, we argue that if the α1-isoform is more significant than the α2-isoform in terms of regulation of contractility, the effects on cardiac contractility would be the inverse of the observed results in the current study. This is further elaborated in the second paragraph of the discussion section (lines 404-422 in the revised manuscript). To accommodate the reviewer’s comment, we also re-phrased the sentence (line 119-120 in the revised manuscript): “This study demonstrates the significant role of the α2-isoform in the regulation of cardiac contractility in a murine model relevant to human disease, i.e., FHM2”.

  • Methods. How did you assess possible effects of ouabain on coronaries, did you observe changes in perfusion pressure?

We understand the reviewer’s concern and agree that this is an important factor for a whole-body physiological response to ouabain. This is, however, out of the scope for this study focused on cardiac muscle function. Thus, we did not aim to assess the coronary function directly, e.g., flow or intraluminal pressure of coronary arteries in the current study.

The perfusion pressure of the system was determined by the water column (hydrostatic pressure) of the perfusion system corresponding to a pressure of 60 mmHg and was constant throughout.

Furthermore, we have discussed this limitation in the Discussion (lines 531-536 in the revised manuscript): “The coronary function of the two genotypes was not assessed in the current study. The prefusion pressure was set to 60 mmHg and maintained throughout the experiment. We cannot exclude differences in coronary flow, however, if the function of the coronary circulation is similar between the two genotypes, the diastolic function may not be sufficiently promoted to compensate for the larger systolic myocardial oxygen consumption in the α2+/G301R hearts.” Notably, this statement is speculation based on the baseline rate-dependent changes (in the absence of ouabain).

  • Methods, statistics: Please describe n. Does it represent biological replicates?

‘n’ represents the number of animals per dataset. As requested, this has now been clarified in the methods section (line 263 in the revised manuscript): “P-values below 0.05 were considered statistically significant, and n indicates the number of animals per dataset”.

  • Results, p.8, para1: When comparing the set end-diastolic pressure, may be an equivalence test would have been more appropriate?

We appreciate the reviewer’s suggestion, and we agree with the reviewer that this is indeed an option. We believe that both the equivalence test and hypothesis testing with a 0.05 threshold for significance are applicable, but the choice of the statistical method depends on the research question. We were not as concerned with the exact range of the end-diastolic pressure, but we questioned whether the end-diastolic pressures were different from each other in the two groups. If there was a difference, the interpretation of the other functional data would be more troublesome.

In contrast, if we were mainly concerned about the end-diastolic pressures residing within our pre-defined range for end-diastolic pressure, we believe an equivalence test would be superior. As an example of this calculation, the 95% confidence intervals for end-diastolic pressure are [7.548-8.775] and [7.811-9.862] for wild-type and α2+/G301R hearts, respectively. These intervals are within our pre-defined range defined in the Methods (line 154 in the revised manuscript). Therefore, for the sake of readability and to not burden the reader with too many technicalities, we believe that the use of a t-test is sufficient and appropriate in the current comparison. To indicate this more clearly, we have changed the phrasing of the sentence (lines 279-280 in the revised manuscript). Additionally, a typing mistake in the main text was discovered (line 279-280 in the revised manuscript), and it has been corrected accordingly to: “The set end-diastolic pressure (WT; 8.2 ± 0.3 mmHg vs. α2+/G301R; 8.8 ± 0.5 mmHg; P = 0.2939) was not different between WT and α2+/G301R hearts.” No similar mistakes were found for the rest of the datasets.

  • Results, p.8, para3: It makes no sense to have a similar cycle duration with longer systolic duration and non-different diastolic duration as systolic duration + diastolic duration = cycle length. Given the variability in diastolic duration and cycle length, did you perform a sample size calculation for these parameters?

The main outcome of this study was the functional variables regarding cardiac contractility. Therefore, the protocol was not designed around the cycle duration and its components. We agree that the discrepancy between changes in the systolic duration and the lack of differences in the diastolic and heart cycle durations is confusing. This is probably the result of a single data point outlier, as both these durations demonstrated a tendency towards prolongation (P = 0.3943 and P = 0.2590, respectively) in α2+/G301R hearts, which is in accordance with the changes in the systolic duration. Our follow-up power analysis indicated that with the current standard deviation of the data, we will need 100 and 50 animals to achieve a significance level of 0.05 with a power of 80% to detect a difference between wild-type and α2+/G301R groups for the diastolic and heart cycle durations, respectively. Because these variables were not the main study objectives, and since they are not important for the main conclusion, they have now been removed from all relevant figures in the revised manuscript to reduce the confusion as indicated by the reviewer.

  • Results, Fig. 7c: In the legend it is stated that “The difference between left ventricular dP/dtmax (LVdP/dtmax) were similar between the genotypes.” but this difference is not shown. Is there a significant difference between the genotypes in the absence and in the presence of ouabain as shown in Fig. 3?

We agree with the reviewer that this was poorly phrased, and it has now been changed accordingly. The two-way repeated measures ANOVA yielded a general genotype effect (P = 0.0370), however following post hoc analysis, the difference between the genotypes without ouabain and in the presence of ouabain did not achieve statistical significance. As was stated in the main body text (lines 319-322 in the revised manuscript), the sentence in the legend has been rephrased to: “The differences in left ventricular dP/dtmax (LV dP/dtmax) between the genotypes did not achieve statistical significance in the absence (P = 0.0652) and in the presence of ouabain (P = 0.0513)”. Furthermore, this has now been indicated clearly in the Figure (Figure 5 in the revised manuscript).

  • Results, Fig. 7f: The statement “the systolic pressure-time index (SPTI) remained larger for α2+/G301R hearts compared to WT hearts” is unclear as no significant difference is indicated in contrast to Fig. 3.

Initially, in Figure 7, we displayed the results of the post hoc analysis only. The difference between the genotypes was reported as a general effect yielded by the two-way ANOVA analysis (P = 0.0181) in the main body text. To clarify this in the figure legend, ‘+’ has been added to the figure to illustrate the difference between SPTI in the absence and in the presence of ouabain. The post hoc analysis only included comparisons of intragroup variables in the absence and in the presence of ouabain. A similar, separate, post hoc analysis of intergroup variables yields the following P-values (P = 0.0431 and P = 0.0152 for WT vs. α2+/G301R in the absence of ouabain and WT vs. α2+/G301R in the presence of ouabain, respectively). Because we believe, this is an important observation, i.e., the increase in SPTI for α2+/G301R hearts while remaining larger compared to WT hearts, we wanted to highlight it. Thus, we believe the addition of ‘+’ to indicate the intragroup effect from the initial two-way ANOVA for α2+/G301R hearts is a fine compromise simplifying and clarifying the figures in conjunction with supporting their interpretation without crowding the figures with too many symbols.

  • Results, p.12, para1: Here you state, for example: “After atrial pacing, the difference in left ventricular dP/dtmax between genotypes did not achieve statistical significance during this period (P = 0.0652 without ouabain and P = 0.0513 in the presence of ouabain; Figure 7c).” It seems confusing the use statistics with different rigor when comparing data shown in Fig. 3-6 and in Fig. 7-10. Did you perform sample size calculations for the data shown in Fig. 7-10, where probably higher n numbers are required to make statistical inference based on the same effect size as shown in Fig. 3-7?

The use of different statistical approaches has been clarified and described in more detail below (see our comments on point 10 and 11). A post-power calculation indicates that “160 and 18 mice are needed to achieve a difference between the genotypes in the dP/dtmax and systolic pressure-time index (SPTI).

Additionally, the P-values have been indicated in the respective Figures for clarification.

  • Results, Fig.S1, S3, S4: Since many parameters change during time control, the evaluation of the effect of ouabain requires the calculation of the differences (control - ouabain) and (control - time control) and the statistical comparison of these differences. What was seen in time control experiments in mutant mice?

We agree with the reviewer that this is another way of setting up the analysis. However, since many variables are different between WT and α2+/G301R hearts at baseline in the absence of ouabain, this approach does not allow us to identify whether the ouabain-induced changes will normalize and equalize this discrepancy between the WT and the α2+/G301R hearts. This will not be possible to identify with statistics on delta values as they only indicate the degree of directional change in variables. Calculating these delta values based on the already presented data and reanalyzing these with a new analysis is a form of data duplication. Thus, we have refrained from choosing this approach. The strategy for the statistical approach of the current study has been clarified and described in detail below (see comment on point 11).

We comprehend and respect the view of the reviewer, therefore, the phrasing regarding an ouabain effect has been made more accurate and careful throughout the main body text (see lines 119-122; 313-314; 351-359; 369-378; 455-465; 468-471; 553-556; and 570-577 in the revised manuscript).

  • Results, Fig. 9 and 10: Why do you show and compare here only 2 groups, in contrast to the data shown in Fig. 7 and 9? This should be unified.

To clarify, we believe the current statistical approaches are viable to convey the message of the current study for the following reasons:

First, we investigated whether α2+/G301R hearts are more contractile than WT hearts at baseline during sinus rhythm. This knowledge is achieved by analyzing the baseline data (Figure 3 in the revised manuscript) with a simple t-test. To test whether this was a rate-dependent difference in the cardiac phenotype, the atrial pacing data were analyzed in a similar manner (Figure 4 in the revised manuscript).

Second, to investigate the cardiac variables in the absence and in the presence of ouabain, we applied a two-way repeated measures ANOVA followed by a post hoc analysis including multiple comparisons. This approach considers the individual pairing of the data points in the absence and in the presence of ouabain.

Last, we wanted to investigate whether changes to cardiac variables in the presence of ouabain are present during atrial pacing. As the initial atrial pacing data has already been analyzed to conclude rate-dependence of the baseline cardiac phenotype (Figures 3 and 4 in the revised manuscript), we cannot include the same data in another analysis in a two-way repeated measure ANOVA based on the atrial pacing data as this will be seen as data duplication. Thus, we also decided to analyze this data with a t-test to determine the rate-dependence of the changes to cardiac variables in the presence of ouabain.
However, we recognize the limitations of the applied statistics as suggested by the reviewer, and we have changed the phrasing of sentences accordingly (See lines from point 10; see lines see lines 119-122; 313-314; 351-359; 369-378; 455-465; 468-471; 553-556; and 570-577 in the revised manuscript).

  • Results, p.15, para 1: The statement “During atrial pacing at 10.5 Hz (630 BPM), ouabain significantly enhanced the left ventricular developed pressure (P = 0.0319) and the left ventricular dP/dtmax (P = 0.0313) of the α2+/G301R hearts compared to WT hearts” is misleading as there is no comparison between parameters in the absence and presence of ouabain. Please revise. If such a comparison is desired to evaluate the effect of ouabain, the effect of ouabain on the desired parameter should be calculated separately in WT and also in mutant mice and these values compared statistically.

We agree with the reviewer that the phrasing is not entirely accurate. Thus, we have modified the sentence accordingly to: “During atrial pacing at 10.5 Hz (630 BPM) and in the presence of ouabain, the left ventricular developed pressure (P = 0.0319) and the left ventricular dP/dtmax (P = 0.0313) of the α2+/G301R hearts were larger compared with WT hearts (Figure 6a and b)”.

Reviewer 2 Report

In this paper, the author investigated the contribution of the α2-isoform of the Na+,K+-ATPase to the cardiac phenotype of mice with the familial hemiplegic migraine type 2 (FHM2) associated mutation in the α2-isoform. The study found that α2+/G301R hearts had increased contractility during sinus rhythm, which was rate-dependent, and an enhanced inotropic effect of ouabain compared to WT hearts, leading to increased systolic work. These results suggest that the α2-isoform of the Na+,K+-ATPase plays a significant role in regulating cardiac contractility, and that reduced expression of this isoform may contribute to the enhanced contractility observed in α2+/G301R hearts. The author is to be commended for their work. However, further research is needed to strengthen the conclusions drawn from this study.

Major issue:

The major issue with this study is that the molecular and cellular mechanisms behind the observed effects of the α2-isoform of the Na+,K+-ATPase on cardiac contractility remain unclear. To address this issue, the author can read the paper exploring the role of the α2-isoform of the Na+,K+-ATPase in neuroinflammation (https://www.nature.com/articles/s41598-020-71027-5), which may provide new ideas for future research.

Minor issue:

1. Please add the scale bar to the cell pictures.

2. Please modify the grammar mistakes in the paper.

Author Response

We would like to thank the reviewer for their suggestions and comments. The discussion section has been improved according to the reviewers suggestion. Please, see the comments for each point below and in the attached document.

Reviewer 2 - Comments and Suggestions for Authors

In this paper, the author investigated the contribution of the α2-isoform of the Na+,K+-ATPase to the cardiac phenotype of mice with the familial hemiplegic migraine type 2 (FHM2) associated mutation in the α2-isoform. The study found that α2+/G301R hearts had increased contractility during sinus rhythm, which was rate-dependent, and an enhanced inotropic effect of ouabain compared to WT hearts, leading to increased systolic work. These results suggest that the α2-isoform of the Na+,K+-ATPase plays a significant role in regulating cardiac contractility, and that reduced expression of this isoform may contribute to the enhanced contractility observed in α2+/G301R hearts. The author is to be commended for their work. However, further research is needed to strengthen the conclusions drawn from this study.

Major issue:

The major issue with this study is that the molecular and cellular mechanisms behind the observed effects of the α2-isoform of the Na+,K+-ATPase on cardiac contractility remain unclear. To address this issue, the author can read the paper exploring the role of the α2-isoform of the Na+,K+-ATPase in neuroinflammation (https://www.nature.com/articles/s41598-020-71027-5), which may provide new ideas for future research.

We agree with the reviewer that this is a limitation of the current study. To address this limitation, a section in the discussion has been added beginning at line 497 including the abovementioned reference with a discussion about possible involved mechanisms and future research. The following paragraph has been added: “We propose two possible underlying mechanisms that may involve the Na+,K+-ATPase α2-isoform. First, the acute effect of ouabain may be effectuated via the conventional Na+,K+-ATPase/NCX pathway modulating [Ca2+]in [8; 14]. Similarly, the greater baseline contractility in α2+/G301R hearts during sinus rhythm compared with WT hearts may also be explained as a mechanism of this pathway. Specifically, a reduced abundance of the α2-isoform may diminish the NCX activity with consequent elevation of [Ca2+]in. However, previously, no difference in membrane conductance between α2+/G301R and WT cardiomyocytes was found [29]. In conjunction with the protein expression analysis indicating no changes in other ion-transporting proteins, e.g., the NCX, this may suggest that a significant change in intracellular ion homeostasis is unlikely [29]. Thus, the mechanism underlying the greater contractility of α2+/G301R hearts at baseline during sinus rhythm may be mediated by another mechanism, e.g., the Na+,K+-ATPase-associated intracellular signaling [63; 64]. Elaboratively, inhibition of the Na+,K+-ATPase α2-isoform with low-dose ouabain (10 μM) has previously been shown to activate the Src kinase, a non-receptor tyrosine kinase [65]. Src activation has been associated with activation of the ERK1/2 pathway [64; 66]. In cardiomyocytes, activation of this pathway has been associated with increased contractility due to changed Ca2+ handling [64]. In a previous study, we have demonstrated that hearts from aged α2+/G301R mice (> 8 months old) display an increased capacity for Src activation together with an increased expression of Src and ERK1/2 kinases compared with hearts from matching WT mice [29]. An increased capacity for Src activation may lead to cardiac alterations resulting in greater cardiac contractility [64]. Elucidation of these distinct mechanisms underlying the acute enhancement of contractility in α2+/G301R hearts in the presence of ouabain and their larger cardiac contractility at baseline compared with WT hearts remain an important objective for future research.”

Minor issue:

  1. Please add the scale bar to the cell pictures.

We thank the reviewer for highlighting the missing scale bars. These have now been added.

  1. Please modify the grammar mistakes in the paper.

The paper has been revised and mistakes have been corrected accordingly (see revised manuscript).

Round 2

Reviewer 2 Report

The authors have answered all of my questions and the paper has been improved.